# Impact of the National Wetland Park in the Poyang Lake Area on *Oncomelania hupensis,* the Intermediate Host of *Schistoma japonicum*

**DOI:** 10.3390/tropicalmed8040194

**Published:** 2023-03-27

**Authors:** Zhaojun Li, Yusong Wen, Dandan Lin, Fei Hu, Qin Wang, Yinlong Li, Jing Zhang, Kexing Liu, Shizhu Li

**Affiliations:** 1Jiangxi Provincial Institute of Parasitic Diseases, Jiangxi Province Key Laboratory of Schistosomiasis Prevention and Control, Nanchang 330096, China; 2Poyang County Schistosomiasis Control Station, Poyang 333100, China; 3National Institute of Parasitic Diseases, China CDC (Chinese Center for Tropical Diseases Research), Key Laboratory on Parasite and Vector Biology, National Health Commission, WHO Collaborating Centre for Tropical Diseases, National Center for International Research on Tropical Diseases, Ministry of Science and Technology, Shanghai 200025, China

**Keywords:** *Oncomelania*, Poyang Lake, *Schistosoma*, travel, wetland park

## Abstract

In this study, we aimed to understand the influence of ecotourism on the distribution of *Oncomelania hupensis* and to provide a scientific basis for formulating effective snail control methods in tourism development areas. Poyang Lake National Wetland Park was selected as the pilot area, and sampling surveys were conducted based on comprehensive and detailed investigations of all historical and suspected snail environments according to map data to determine the snail distribution and analyze the impact of tourism development. The results showed that from 2011 to 2021, the positive rates of blood tests and fecal tests tended to decrease among residents of the Poyang Lake area. The positive rates of blood tests and fecal tests in livestock also tended to decrease. The average density of *O. hupensis* snails decreased, and no schistosomes were detected during infection monitoring in Poyang Lake. The local economy rapidly grew after the development of tourism. The development of ecotourism projects in Poyang Lake National Wetland Park increased the transfer frequency of boats, recreational equipment, and people, but it did not increase the risk of schistosomiasis transmission or the spread of *O. hupensis* snails. Prevention and monitoring only need to be strengthened in low-endemic schistosomiasis areas to effectively promote economic development due to tourism activities without affecting the health of residents.

## 1. Introduction

Schistosomiasis is a parasitic disease that seriously endangers human health and affects social and economic development. Human schistosomes mainly include *Schistosoma japonicum*, *Schistosoma mansoni*, *Schistosoma haematobium*, *Schistosoma intercalatum*, *Schistosoma mekongi*, and *Schistosoma malayensis*, where the first three are the most common in humans. The characteristics and geographical distributions of the intermediate host snails also vary for different schistosome species [1,2]. *Oncomelania hupensis* is the only intermediate host of *S. japonicum*, which is mainly distributed in China, Japan, the Philippines, and Indonesia.

The distribution of *O. hupensis* is consistent with the endemic schistosomiasis areas. In China, *O. hupensis* is mainly distributed in 12 provinces (municipalities and autonomous regions) in southern China. After more than 60 years of comprehensive prevention and control measures, China has made great achievements in the prevention and control of schistosomiasis to reach transmission control, and this disease is now at a low epidemic level. The control of snails played a key role during this period. According to statistics at the end of 2021, the actual area affected by snails in China was 369,268.74 ha, which is a decrease of 74.21% compared with the historical snail-affected area of 1,432,100 ha [3].

Environmental changes can severely affect the growth, reproduction, and diffusion of snails, but they also affect the prevalence and transmission of schistosomiasis [4,5,6]. Rapid social and economic development accompanied by the construction of many major projects have led to drastic changes in the natural and ecological environment, which could influence the distribution of snails [7,8,9]. Long-term prevention and control programs have shown that changes in the ecological environment may be beneficial for the control or elimination of schistosomiasis, but they may also lead to the spread of snails and schistosomiasis [10,11,12]. There is a potential risk of schistosomiasis epidemics in areas where construction projects are implemented, and thus controlling the schistosomiasis situation is still very important [13,14,15]. Studying the effects of environmental change on the prevention and control of schistosomiasis is important for sustainable social and economic development in China [16,17,18].

Jiangxi Province is an area of China with a very high prevalence of schistosomiasis. The schistosomiasis epidemic areas are mainly concentrated in 17 counties (cities and districts) along Poyang Lake, which has been the main problem area for schistosomiasis control in Jiangxi Province [19]. Ecological tourism and other economic development measures have brought new challenges for local schistosomiasis control and snail control. The spread of snails and the transmission of schistosomiasis can only be effectively prevented by considering schistosomiasis transmission control in ecological construction and by formulating corresponding control measures [20].

Poyang Lake is located in the north of Jiangxi Province, and it is the largest freshwater lake in China with clear water and a beautiful landscape. Poyang County is the county located most closely to Poyang Lake in Poyang Lake Ecological Economic Zone. The county has a total area of 4215 square kilometers, and it currently governs 29 townships, two reservoir administrations, 476 administrative villages, and a total population of 1.56 million. The largest county in the province is also a well-known habitat for migratory birds from around the world. More than 95% of white cranes in the world fly thousands of miles to Poyang Lake in the cold winter. Poyang County has a unique advantage in terms of eco-tourism. Extensive wading projects and fish custom cultural experience activities have been implemented as important tourist attractions. The number of tourists that visited the area during 2019 was 200,000 [21].

The Poyang Lake area has been a schistosomiasis endemic area for many years. A comprehensive prevention and control strategy based on controlling the source of infection has been implemented in recent years with remarkable effects [22,23], but the risk of schistosomiasis transmission is still significantly higher in the Poyang Lake area than in other lake areas [24,25]. This problem is due to the vast size of the Poyang Lake area, insufficient grassland management, and resource development and utilization. The investment in controlling the source of infection is large, but it cannot generate direct economic benefits, thereby conflicting with the interests of ordinary people in the affected areas [26]. The terrain is complex in the Poyang Lake area, and many different types of wild animals are present. Wild animals infected with schistosomiasis are potential sources of infection. Therefore, the development of tourist attractions and projects poses a serious threat to the health of local residents, tourists, and tourism practitioners [27,28].

Zhu Lake is located in the middle of Poyang County at the east edge of Poyang Lake. Seven townships are present along Poyang Lake, comprising Zhu Lake, Baishazhou, Zhanqian, Sishili Street, Tuanlin, Shuanggang, and Niejia. Many ports are also present in the lake, which has “48 major branches” and “84 small branches”. The surface area of the lake is 4100 ha, and the average water depth is 4 m, with a maximum depth of over 10 m.

Eight points were selected for water body monitoring in this study, i.e., both sides of Poyang Lake Zhuhu flood discharge gate (outer lake), Poyang Lake Tourist Wharf, Xiaoleijiao, Shanggouzi Lake Island, Xigangping, Xishanping, and Xiangyouzhou Wharf and Bird Watching Platform, and four other points were also selected, comprising the tourist center pier in Zhu Lake, Zhu Lake Wharf, Zhu Lake Floodgate (inner lake) gate, and Douli Mountain.

## 2. Materials and Methods

### 2.1. Overview of the Study Area

The planning and construction of Poyang Lake National Wetland Park started in 2008, and it entered into operation in 2015. Poyang Lake Wetland Park is located in Poyang County, Jiangxi Province, China, at the confluence of the Raohe and Zhangtian rivers on the east bank of Poyang Lake (116°23′39″–116°44′38″ E, 28°56′52″–29°13′31″ N). The area of the wetland park includes the water surface and shoal of Poyang Lake, the water surface of Zhu Lake, Qingshan Lake, Tu Lake, East Lake, Lean River, and Changjiang River within the administrative jurisdiction of Poyang County, as well as a parcel measuring 1.8 km long and 1 km wide between Qingshan Lake and Zhu Lake. The planned total area of the park is 36,285 ha, and the total area of the wetland park is 35,116.1 ha, which accounts for 96.8% of the total area. Poyang Lake National Wetland Park is one of the six largest wetlands in the world, with the greatest abundance of wetland species in Asia.

### 2.2. Scope of Investigation

The Poyang Lake National Wetland Park tourism project includes 11 grasslands comprising Xiangyouzhou, Yihuangping, Changshanping, Xishanping, Xigangping, Xiaoleijiao, Wugangping, Pangpizhou, Baijiazui, Chenjiacha, and Zhanjialong.

Systematic sampling was applied to conduct on-site snail surveys. The linear distance covered in each grassland was 30–50 m; the point distance was 10 m; and each frame measured 0.1 m^2^. To perform the environmental sampling survey, all historical snail environments or suspected snail environments, such as grasslands, lake branches, beaches, and embankments in the water of Zhu Lake, were investigated by setting up separate frames according to the map data. The investigation focused on suspected areas with a history of snail occupation; complex bifurcated lake environments; and the rivers, ditches, and canals connected to Zhu Lake.

Eight administrative villages (Rongqi Village, Longtoushan Village, Niejia Village, Lehu Village, Changshan Village, Wangjia Village, Huzhao Village, and Chemen Village) near Zhu Lake and the Poyang Lake project area were selected to investigate the schistosomiasis infection status in humans and animals.

### 2.3. Snail Survey

Retrospective studies, field investigations, epidemiological research methods, and Geographic Information System (GIS) tools were used to investigate suspected snail breeding environments in Zhu Lake (inner lake) and Poyang Lake (outer lake) in the wetland park project. In all of the investigations, crushing was used to distinguish dead snails and infectious snails. The density of infected snails, the infection rate, and the incidence of rimmed snails were calculated.

### 2.4. Investigation on Infection Situation of Population and Livestock

A retrospective investigation was conducted based on epidemic data for determining the current epidemic status. In each village, 300-to-500 villagers aged 6-to-65 years were selected as clusters and screened using the indirect hemagglutination (IHA) test. People who were positive for IHA were then tested using the Kato–Katz method (three tests for each sample). Samples with schistosome eggs were confirmed cases.

At the same time as we investigated the human population, 50 cows were selected from each of the eight administrative villages mentioned above (if the number of cattle was less than 50, all were inspected) and the feces were collected to assess schistosome egg hatching (three tests for each sample). Animals with *Schistosoma miracidium* were confirmed as sick animals.

### 2.5. Water Monitoring

Each measurement point was divided into two groups (two mouse measurement cages), and each group (one measurement cage) contained 20 mice (4 mice per grid). The two groups were dragged and dropped at a distance of 50 m and suspended on the water’s surface. When dragging and dropping, we ensured that the limbs and abdomen of each sentinel mouse were in contact with the water body. Measurements were conducted for 2 h in the morning and afternoon every day for two consecutive days, for a total of 8 h.

After feeding the mice for 35–40 days (if the mouse died after 28 days of feeding, it was also dissected), the mice were dissected and carefully observed to determine whether adult schistosomes were present in the hepatic portal vein and mesenteric vein. All *Schistosoma japonicum* adults were observed under a dissecting microscope to distinguish male and female worms, and to count them.

### 2.6. Data Analysis

All data obtained in each part of the study were collated and statistically analyzed with Microsoft Excel.

## 3. Results

### 3.1. Distribution of Snails

The distribution area in the Poyang Lake National Wetland Park tourism project comprises the inner lake and outer lake.

The inner lake is Zhu Lake, which maintains a water elevation level of 12–16 m (Yellow Sea elevation) throughout the year. The water surface area is 5990 ha; the shoreline, about 140 km long and 1–40 m wide; and the water level fluctuation zone, about 100 ha. The lake banks comprise sections with steep cliffs, branching meadows, gentle lake beaches, and artificial dikes. Shuanggang, Sishili Street, Gaojialing, Zhu Lake, Baishazhou, and other towns are the main areas for water tourism and sightseeing projects in Poyang Lake National Wetland Park.

The outer lake is Poyang Lake, and its water level rises and falls. This is the main site for yachting, migratory bird watching, karting, and wetland sightseeing in Poyang Lake National Wetland Park. The core area of the project is Xiangyouzhou, where bird watching platforms, karting, wetland sightseeing, and other projects are surrounded by Yihuangping, Changshanping, Xishanping, Xigangping, Xiaoleijiao, Wugangping, Pangpizhou, Baijiazui, Chenjiacha, and Zhanjialong, and 10 other grasslands. The project has an area of 7336 ha, with a historical snail-affected area of 1423 ha.

Zhu Lake and Poyang Lake are separated by the Zhu Lake levee, which maintains a relatively stable water level in the inner lake. In normal years, during the dry season, grassland is exposed in Poyang Lake, and the water level of the inner lake is generally higher than that of the outer lake. This period is relatively long and usually from September to May in the following year. During the flood season, grassland is submerged under water in Poyang Lake, and the water level of the outer lake is generally higher than that of the inner lake. This period is relatively short and usually from June to September every year. The Zhu Lake floodgate connects the inner lake with the outer lake. The inner-lake water elevation level can be controlled to 14–18 m by adjusting the closure of a five-hole gate.

### 3.2. Snail Survey

The vegetation on the high shore of Zhu Lake is mainly dominated by *Hemarthria compressa*, and the vegetation on the low beach and mudflat mainly comprises *Carex*. *H. compressa* is not suitable for snail breeding, and the coverage of sedge is low on the beach and mudflat. The flooding time is long, so this area is unlikely to be suitable for snail breeding. 

In total, 114 environments were investigated along the 140 km long lake and the 1–40 m wide lake shorelines, backwaters, beaches, and polders. The total area investigated was about 156 ha. In total, 8960 frames were investigated, and no snails were found in four consecutive years.

The 10 grasslands is the core tourism area of Poyang Lake, and snail surveys showed that all of the grasslands in the project area had snail-affected beaches. Positive snail de-tection results were obtained each year from 2006 to 2009, but not in 2010 and 2011 (Table 1).

The number of snails gradually decreased after 2012. After 2012, the number of grasslands with snails gradually decreased. Snails were found in two grasslands in 2015, and the average density of snails was only 0.00025/0.1 m^2^. In 2017, only nine snails were found on 1500 ha of grassland, and a few snails were found in Xiangyouzhou, Changshanping, and Zhanjialong from 2016 to 2017. In 2018, snails were found in Xishanping, but the density was relatively low. The survey results from 2011 to 2021 show that the density of snails fluctuated, but the variations were small (Figure 1).

### 3.3. Schistosomiasis Infections in Wetland Park Population

Eight endemic schistosomiasis villages were selected for investigation in Zhu Lake Township, Rongqi Village; Shuanggang Town, Longtoushan Village; Nijia Village, Lehu Village, Changshan Village; Baishazhou Township, Wangjia Village; Huzhao Village; and Chemen Village. From 2004 to 2008, acute schistosomiasis occurred in all eight villages for four years, and 10 acute patients (students) were infected while swimming in the water. Four grasslands were investigated in Poyang Lake, comprising Changshanping, Zhanjialong, Pangpizhou, and Chenjiachaping. Schistosomiasis infections were surveyed in eight villages from 2011 to 2021. The history of positive blood test cases declined, and the fecal test results were negative after 2013 (Figure 2).

### 3.4. Bovine Infections with Schistosoma japonicum 

The number of cattle decreased from 1081 in 2011 to 246 in 2016 due to measures such as forbidding the depasturage of livestock on marshlands. Cattle in the eight endemic schistosomiasis villages grazed on grassland in Poyang Lake before 2011, and the average fecal positive rate was 22.22%. After 2012, the number of farm cattle and schistosomiasis infection rate significantly decreased due to machine cultivation and the forbidding of the depasturage of livestock on marshlands, and all the stool tests were negative after 2013 (Table 2).

### 3.5. Water Infectivity Monitoring

From 2011 to 2018, 40-to-80 mice were released to monitor the infectivity of water, and none were infected with schistosomes. No schistosomes were detected at entrance No. 2 of Zhu Lake and the gate of Baishazhou (Table 3).

### 3.6. Economic Trend

Due to the strong economic development of Poyang Lake National Wetland Park, the core area of the scenic spot comprising Baishazhou Township has been rejuvenated by tourism, and the economy has greatly developed. Fishermen have changed their jobs to participate in the development of tourism. The number of people engaged in “agri-tainment” in Baishazhou Township increased from 2 to more than 60, and more than 300 villagers have been lifted out of poverty to a much wealthier status. In order to strengthen the industrial support for the construction of the ecological fishing village, the aquatic tourism resources of Poyang Lake should be rationally developed. It is necessary to expand and strengthen the aquatic product brand based on the platform of Poyang Lake National Wetland Park. The township could further develop the tourism market in the fishing village by focusing on the characteristics of the lake area in a “one village, one product” project by emphasizing the regional culture, human traditions, farm life, and green catering. Villagers should be encouraged to engage in a various business activities, such as agri-tainment, fish fun houses, local products, and craft sales. The industrial structure could be adjusted to accelerate the breeding and promotion of whitebait, turtles, and river crab. In addition, improved awareness of the aquatic product brand is also an option. As a consequence, ecological tourism and agricultural development can be integrated in an organic manner to make Baishazhou into a beautiful scenic spot for aquaculture, leisure, and tourism. The township receives more than 30,000 tourists every year and the income from tourism increases each year. In 2018, the total income from tourism was CNY 2.166 million. In 2019, Baishazhou Township had 10 stores/supermarkets, with a business area of more than 50 m^2^ [29].

## 4. Discussion

Efforts to control *O. hupensis* are strong in China but reappearances and new occurrences still occur in some areas; thus, the breeding area for *O. hupensis* has been around 3.6 billion m^2^ for many years [30]. The presence of schistosomiasis-infected individual wild mammals poses a significant potential risk to the integrity of the schistosomiasis transmission chain [31,32].

*O. hupensis* control is facing many challenges due to various main factors, such as reductions in the comprehensive management of *O. hupensis* control projects and environmental protection requirements [33,34].

The construction of Poyang Lake National Wetland Park has resulted in significant ecological, social, and economic benefits. The wetland park is expected to effectively protect the integrity of the wetland ecosystem of Poyang Lake, protect the typical lake wetland landscape close to nature, protect and improve the habitats of wetland organisms, protect and restore biodiversity, and give full play to the roles of Poyang Lake wetland in water conservation and flood regulation, in purifying pollutants, in adjusting the climate and gas levels, in leisure and entertainment, and in cultural research, as well as other functions. In addition, the construction of Poyang Lake National Wetland Park has significantly increased the popularity of Poyang County by further optimizing the investment environment; achieving sustainable, high-speed, and stable social and economic development; and forming a relatively strong wetland cultural atmosphere to effectively satisfy the growing needs of people. The increasing material and cultural needs of life, beside improving the community’s awareness of ecological environment protection and wetland protection, have important practical significance for building a harmonious society in Poyang County and constructing a new socialist countryside. For example, Hunan Province included the prevention and control of schistosomiasis in the overall plan for the development of local social undertakings, as well as incorporating the prevention and control of schistosomiasis in “Dongting Lake Ecological Economic Zone Planning” and “Three-Year Action Plan for Special Improvement of the Dongting Lake Ecological Environment (2018–2020)”, thereby providing a strong guarantee for schistosomiasis prevention and control work in the province [35]. The results obtained in the present study demonstrate that the integrated environmental improvement of marshlands by implementing industrial, agricultural, and water resource development projects can alter snail habitats in marshland regions and promote local economic development, which appears to be a win–win strategy for blocking the transmission of *S. japonicum* and accelerating socio-economic development along Yangtze River [36].

*O. hupensis* is the only intermediate host of schistosomiasis, and the geographical distribution of this snail affects the distribution of endemic schistosomiasis [37]. By monitoring the spatial distribution and number of *O. hupensis*, we determined the endemic area for schistosomiasis [38]. The climate, soil, plants, and other natural conditions in Poyang Lake are highly suitable for snail survival. The water level of Poyang Lake has greatly fluctuated due to global warming, but there has been little change in the snail distribution in Poyang Lake [39]. The prevalence of schistosomiasis is low, but the schistosomiasis transmission chain has not been broken, and the risk of transmission still exists [40]. Therefore, it is very important to monitor snails in Poyang Lake National Wetland Park during tourism activities.

Our results showed that the distribution of snails in Poyang Lake was different from that in the outer lake. *O. hupensis* was found in the outer lake but not in the inner lake. The density of *O. hupensis* was not high in the inner lake and far lower than the average live snail density of 0.010 per 0.1 m^2^ in the national monitoring point at Poyang County, but the annual live snail density fluctuates to some extent [41]. Therefore, it is necessary to strengthen the monitoring and control of snails in the outer lake of Poyang Lake National Wetland Park to prevent the spread of snails and the transmission of schistosomiasis.

In the years after the construction of the polder, the vegetation in the inner lake will likely change, and the snails will likely die out naturally [42]. After the Zhu Lake levee was built in 1975, the ecological environment changed for snails in the levee, and it became a snail-free area after 3 years. This is consistent with the absence of snails in the polder three years after the construction of the dikes in Poyang Lake. In total, 5960 frames were investigated in the environment along the 140 km shoreline of Zhu Lake, but no snails were found. Thus, after the construction of the dike, the water level of Zhu Lake increased and became relatively stable, and the tidal flats along the lake were submerged for many years; the area subsequently became snail-free. However, Zhu Lake and Poyang Lake are separated by only one dike. Zhu Lake used to be an area with a history of snails, and its ecological environment has not fundamentally changed in recent years. Therefore, the waterline area may still be suitable for snail breeding. Due to the vigorous development of ecotourism projects in the region, increases have occurred in the frequency of boats, other recreational equipment, and tourists flowing between Poyang Lake and outside areas; thus, the possibility of snails or snail eggs attached to cruise boats entering Poyang Lake has increased. In addition, the use of fishing boats and fishing tools by fishermen near the wetland park in Zhu Lake and Poyang Lake increases the possibility of snails or snail eggs becoming attached to fishing gear; thus, they may enter Zhu Lake from Poyang Lake. Snails are still distributed in the upper reaches of the river channel connected to Zhu Lake, and they may spread to the lower reaches of Zhu Lake under flooding. These risk factors may allow snails to spread to the Zhu Lake area.

From 2011 to 2021, a survey of schistosomiasis infections among residents in eight villages in Poyang Lake National Wetland Park showed that the number of blood positive cases decreased, and the fecal test results were all negative after 2013. Schistosomiasis control has been effective in Poyang Lake Wetland Park, but schistosomiasis has not been eliminated, and there is still a risk of transmission. Therefore, it is necessary to strengthen schistosomiasis detection and treatment, as well as health education for staff and visitors in Poyang Lake Wetland Park. The development of tourism has increased the number of jobs for local residents that do not come into contact with infected water, and avoiding schistosomiasis is beneficial to their health.

Water infectivity testing using mice in Zhu Lake found no evidence of schistosomiasis infections. However, Zhu Lake is connected to Poyang Lake through a sluice gate, and the design of the sluice gate is conducive to preventing snails from spreading from Poyang Lake to Zhu Lake. The water flow is discharged from Zhu Lake to Poyang Lake during the “storage period” and the “discharge period”. Therefore, there is no risk of snails spreading to Zhu Lake during the “storage period” and the “discharge period”. Water flows from Poyang Lake to replenish the water in Zhu Lake in the “water replenishment period”, but the operating rate is very low in the “water replenishment period”. In addition, the culvert pipe is designed to be high at one end of Zhu Lake and low at the other end of Poyang Lake, so snails floating with the current would even have difficulty entering Zhu Lake through the culvert pipe. Moreover, if snails at the bottom entered the lake with the current, it would be difficult for them to survive, because the lake is filled with water all year round. Therefore, the possibility of snails spreading from Poyang Lake to Zhu Lake through the sluice gate is very small. However, although the risk of snail diffusion is very small, the prevalence of schistosomiasis in the outer lake is still severe. During the “water replenishment period”, the outer lake may carry *schistosome cercariae* into the inner lake, which may lead to the infection of humans or livestock in the water on the lake side of the gate. The number of years when water replenishment is required is decreasing due to the changing hydrological conditions, but the risk of schistosomiasis cannot be ignored in some locations. Long-term snail monitoring is essential in wetland areas.

## 5. Conclusions

Ecotourism in the Poyang Lake region has a direct effect on the transmission of schistosomiasis, with advantages and disadvantages. In order to ensure the sustainable and healthy development of the wetland park, strategies and measures to prevent schistosomiasis have been strengthened, such as preventing the spread of *O. hupensis* and controlling the source of infection. Fishermen actively have moved onshore to engage in tourism-related industries, which reduced the risk of schistosomiasis infection and increased the economic income. A large number of tourists may be exposed to infected water, which can spread schistosomiasis. The transmission of schistosomiasis will only be effectively blocked when the control of schistosomiasis transmission will be considered in ecological construction, and corresponding control measures will be formulated. Therefore, it is necessary to establish surveillance and control measures to prevent the risk of schistosomiasis transmission in such areas, so that economic development and disease control can be promoted in schistosomiasis endemic areas, with complementary effects on each other. According to snail investigations, field epidemiological investigations, and historical analysis of schistosomiasis in Poyang Lake National Wetland Park, the development of the ecotourism industry has greatly contributed to economic growth for the local residents, as well as creating a good environment for residents to reduce their exposure to schistosomiasis via water and promoting the local control of schistosomiasis.

## Figures and Tables

**Figure 1 tropicalmed-08-00194-f001:**
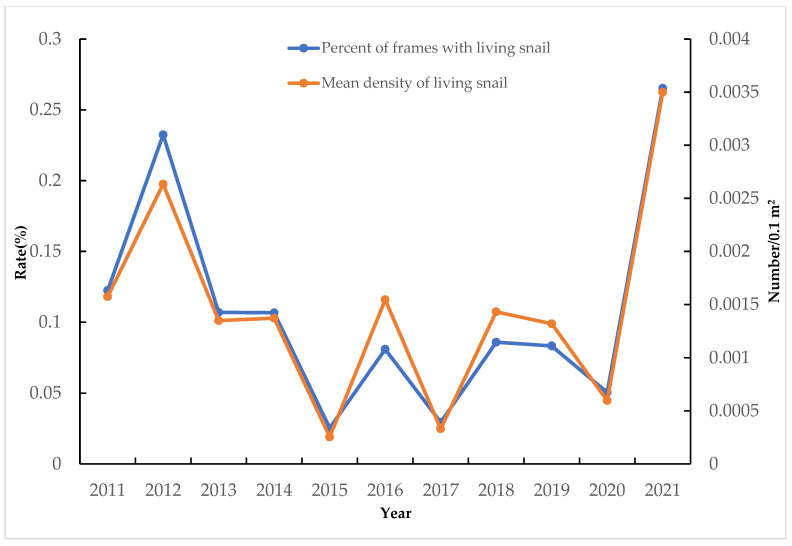
Distribution of *Oncomelania hupensis* in the outer lake of the wetland park. The 95%CI of the percent of frames with living snail: [0.056710589321, 0.159512460822], and the 95%CI of the mean density of living snails: [0.00080622112268, 0.00208564245935].

**Figure 2 tropicalmed-08-00194-f002:**
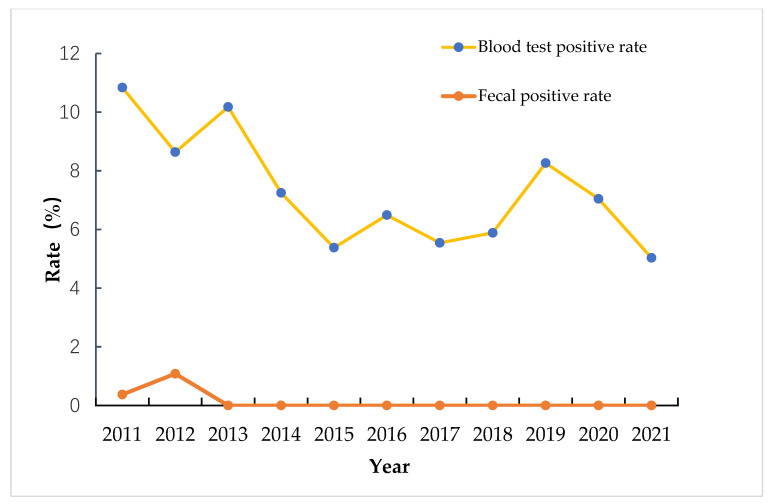
Schistosomiasis infections among people in wetland park areas. The 95%CI of the blood test positive rate: [6.01048825683718, 8.63441466482174].

**Table 1 tropicalmed-08-00194-t001:** Snail situation in the outer lake of Poyang Lake National Wetland Park from 2006 to 2011.

Year	Investigated Snail Area (ha)	Number of Snail Frames (Box)	Number of Live Snail Frames (Box)	Number of Live Snails (Pcs)	Number of Positive Snails (Pcs)	Rate of Snail Frame Occurrence (%)	Mean Density of Living Snails (Pcs/0.1 m^2^)	Positive Snail Density (Pcs/0.1 m^2^)	Area of Positive Snails (ha)
2006	1187.99	21,535	744	1434	25	3.46	0.0666	0.0012	25.0
2007	322.75	7430	528	1034	6	7.11	0.1392	0.0008	6.0
2008	531.54	10,965	868	1392	12	7.92	0.1269	0.0011	12.0
2009	514.65	8052	444	589	9	5.51	0.0731	0.0011	9.0
2010	644.07	10,586	213	355	0	2.01	0.0335	0.0000	0.0
2011	1132.41	25,458	35	43	0	0.14	0.0017	0.0000	0.0
Total	4333.41	84,026	2832	4847	52	3.37	0.0577	0.0006	52.0

**Table 2 tropicalmed-08-00194-t002:** Infection of schistosomiasis among farm cattle in Wetland Park.

Year	Number of Cattle Inspected	Number of Positive Blood Tests	Rate of Positive Blood Tests (%)	Number of Positive Stool Tests	Rate of Positive Stool Tests (%)
2011	1081	44	4.07	23	2.13
2012	1801	55	3.05	26	1.44
2013	563	9	1.60	4	0.71
2014	483	3	0.62	0	0
2015	385	2	0.52	0	0
2016	246	1	0.41	0	0
2017	0	0	0	0	0
2018	0	0	0	0	0

**Table 3 tropicalmed-08-00194-t003:** Monitoring of schistosomiasis in wetland park water.

Year	Monitoring Location	Longitude (E)	Latitude (N)	Number of Mice Released	Number of Mice Recovered	Number of Dissected Mice	Number of Positive Mice
2011	Luodun	116.57780	29.11890	40	34	34	0
2012	Luodun	116.57780	29.11890	40	38	38	0
2013	Luodun	116.57780	29.11890	40	39	39	0
2014	Luodun	116.57780	29.11890	40	40	40	0
2015	Baijiazui	116.36611	29.09514	24	24	24	0
	Luodun	116.57780	29.11890	22	22	22	0
2016	Baijiazui	116.36611	29.09514	40	37	37	0
	Chenmen Zakou	116.61813	29.16068	10	10	10	0
	Neihu Matou	116.60868	29.15679	10	10	10	0
	Luodun	116.57780	29.11890	40	40	40	0
2017	Baijiazui	116.36611	29.09514	20	20	20	0
	Chenmen Zakou	116.61813	29.16068	40	40	40	0
	Neihu Matou	116.60868	29.15679	18	18	18	0
	Luodun	116.57780	29.11890	40	39	39	0
2018	Baijiazui	116.36611	29.09514	20	18	15	0
	Chenmen Zhakou	116.61813	29.16068	20	20	18	0
	Neihu Matou	116.60868	29.15679	20	19	19	0
	Luodun	116.57780	29.11890	20	19	19	0

## Data Availability

Not applicable.

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
