# Peer review of "Impact of the National Wetland Park in the Poyang Lake Area on Oncomelania hupensis, the Intermediate Host of Schistoma japonicum"

_tropicalmed, 2023, doi:10.3390/tropicalmed8040194_

Round 1

Reviewer 1 Report

Dear authors,

Thank you for this notable research work. Below are the comments and suggestions on the research paper entitled “Impact of the National Wetland Park in the Poyang Lake area on Oncomelania hupensis, the intermediate host of Schistosoma japonicum".

The manuscript is well written and has presented significant points on the effect of the National Wetland Park on the presence and transmission of S. japonicum by its snail intermediate host. However, the following comments must be addressed to further improve the research paper.

 Introduction section lines 34 and 35.

§  I would like to clarify the scientific names of schistosomes written in the introduction part.

§  These were the scientific names written. S. mannii, S. egyptiana, S. intercala, S. megong, and S. malei

§  Is S. manni referring to Schistosoma mansoni?

§  Is S. intercala referring to Schistosoma intercalatum?

§  Is S. megong referring to Schistosoma mekongi?

§  How about S. malei and S. egyptiana?

 Section 3.3 Bovine infections with Schistosoma japonicum

§  Line 189: The scientific name Schistosoma japonicum must be italicized in section 3.3.

§  Line 193-195. It says that after 2012, the number of farm cattle and schistosomiasis infection rate decreased significantly due to machine cultivation and forbidding the depasturage of livestock on marshlands, all the stool tests were negative after 2013. Is there reference data or research work for this to substantiate the information provided? 

 Section 3.4 Water infectivity monitoring

§  Line 196-199. Is there reference data for the presence of schistosome infection in mice to substantiate the information provided?

Section 239. The scientific name Oncomelania hupensis must be italicized.

Discussion section line 286. It says that water infectivity testing using mice in Zhu Lake found no evidence of schistosomiasis infections. Is there reference data for this to substantiate the information provided?

Discussion:

Information in the discussion can be improved by providing specific examples of snail control measures and surveillance, especially in the outer lake which poses a risk of the acquisition of schistosome infection. What are the current challenges in the control program? How did the control measures reduce the prevalence of schistosomiasis in many endemic provinces in China? A simple highlight of this information in the discussion would be good to intensify the content presented.

Conclusion:

Line 307. It says that “Ecotourism in Poyang Lake region has a direct effect on the transmission of schistosomiasis, which has both advantages and disadvantages”. Kindly expound on this statement explaining the advantages and disadvantages.

These are some of the suggestions to substantiate the relevance of this research paper. Kindly highlight in red the revisions made in the revised manuscript for confirmation upon submission of the revised paper.

Thank you for your continued research work.

Author Response

We have made supplementary modifications according to the reviewer's opinion. Please see the red text in the attachment. Thank you.

Reviewer 2 Report

The method description is too short. I found several subsections of the result section that were not covered in the methods section, for example, bovine infection, and monitoring with mice. Please extend the methods section and give a more detailed description of all methods used.

For the data presented in the figures please add the confidence intervals or standard deviation for the data points. 

Author Response

(The authors gave the same response as above.)
